# Biofuels from Pyrolysis of Third-Generation Biomass from Household and Garden Waste Composting Bin: Kinetics Analysis

**Bruna Rijo [1], Ana Paula Soares Dias [2],*, Novi Dwi Saksiwi [2], Manuel Francisco Costa Pereira [2], Rodica Zăvoianu [3], Octavian Dumitru Pavel [3], Olga Ferreira [2] and Rui Galhano dos Santos [2]**

[1] CoLAB BIOREF—Collaborative Laboratory for Biorefineries, 4466-901 São Mamede de Infesta, Portugal; bruna-rijo@bioref-colab.pt

[2] CERENA, Instituto Superior Técnico, Universidade de Lisboa, Av. Rovisco Pais, 1, 1049-001 Lisboa, Portugal; novi.saksiwi@tecnico.ulisboa.pt (N.D.S.); mfcp@ist.utl.pt (M.F.C.P.); orferreira@fc.ul.pt (O.F.); rui.galhano@tecnico.ulisboa.pt (R.G.d.S.)

[3] Research Center for Catalysts & Catalytic Processes, Faculty of Chemistry, University of Bucharest, 4-12 Regina Elisabeta Bd., 030018 Bucharest, Romania; rodica.zavoianu@chimie.unibuc.ro (R.Z.); octavian.pavel@chimie.unibuc.ro (O.D.P.)

* Correspondence: apsoares@tecnico.ulisboa.pt

**Abstract:** The modern society produces large amounts of household waste with high organic matter content. The vermicomposting of household waste produces high-value humic substances and is a way to stabilize organic material for later use as raw material (3rd generation biomass) for bioenergy proposes. A 6-month matured compost, combining vegetable and fruit scraps from domestic trash and grass and shrub clippings from yard waste, was evaluated to assess its potential as a raw material in pyrolysis processes. The pyrolysis activation energy (Kissinger) of the composted material showed values in the range of 200–300 kJ/mol, thus confirming its suitability for pyrolysis processes with promising $H_2$ yields. The treatment of the composted material with $H_2SO_4$ and NaOH solution (boiling; 1 mol/L) led to the production of solid residues that present higher pyrolysis activation energies, reaching 550 kJ/mol for the most resilient fraction, which makes them suitable to produce carbonaceous materials (biochar) that will have incorporated the inorganics existing in the original compost (ashes 37.6%). The high content of inorganics would play a chief role during pyrolysis since they act as gasification promoters.

**Keywords:** biofuel; biomass composting; pyrolysis; 3rd generation biomass; thermogravimetry; kinetics

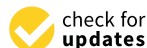



## 1. Introduction

Today's society produces great quantities of food waste that must be addressed and controlled to reduce the environmental impact. The issue of food waste generation affects people of all income levels (Figure 1). In terms of calorie content, it is predicted that worldwide food waste at the consumer level will, when compared to 2013, nearly double by 2050 [1].

Food waste is organic material that decomposes in the environment, generating unpleasant odors and greenhouse gases ($CH_4$, $CO_2$, $N_2O$, and chlorofluorocarbons [2]). Leachates from food waste decomposition can permeate soils and contaminate aquifers. Moreover, the world population will reach 9.8 billion by 2050 (data from the United Nations organization, https://www.un.org/en/desa/world-population-projected-reach-98-billion-2050-and-112-billion-2100 (accessed on 17 May 2023)), and under current conditions, it will be impossible to produce food to feed the entire population. For this reason, finding a solution to food waste is imperative. Reducing food waste is a key factor in mitigating greenhouse gas emissions from agriculture and industries related to food production and distribution [3].

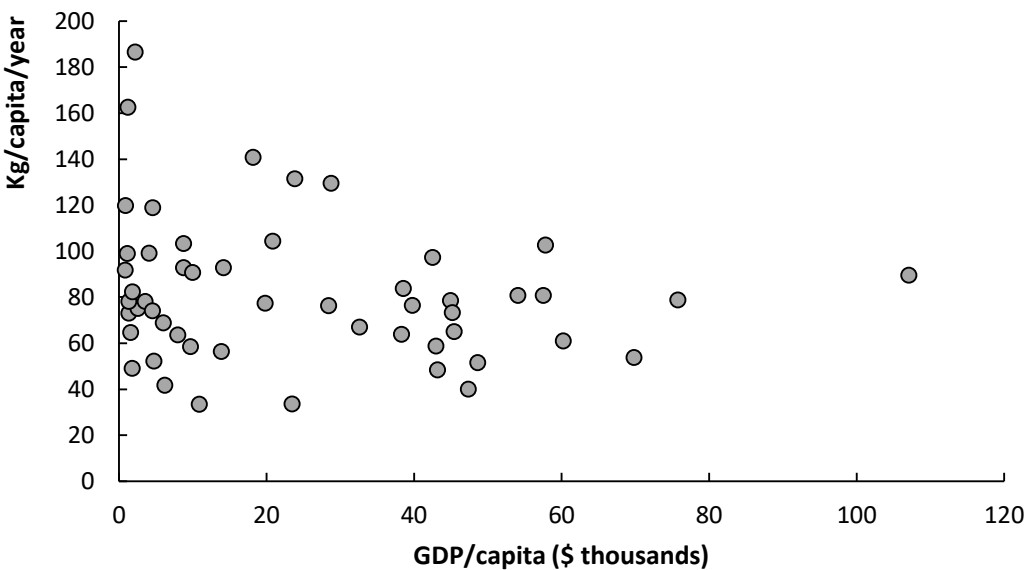

**Figure 1.** Year 2021 food waste versus income (GDP-gross domestic product; adapted from [4]).

Within the framework of the circular economy governed by the 3Rs, reduce, recycle, and reuse, food waste management should follow the hierarchy shown in Figure 2. Nonetheless, the landfill will always be the last alternative for food waste management, and composting should be researched to maximize the environmental benefits connected with it. Composting is a natural process that involves the aerobic biological breakdown of organic wastes (green waste, household waste, sewage sludge, etc.). Composting and vermicomposting are considered green technologies [5]. Composting is the controlled biodecomposition and subsequent stabilization of mixed organic materials such as lignocellulose, proteins, and oils. During composting, cellulose is broken down into cellobiose and glucose (Figure 3), which are consecutively converted into organic acids, $CO_2$, and water. Xylan, the most abundant component of hemicellulose, is converted into its oligomers, which subsequently undergo hydrolysis into organic acids and alcohols, followed by their decomposition into $CO_2$ and water. Lignin, whose content increases with the age of the plant, is the most recalcitrant component in compost, but its total degradation produces natural aromatic compounds such as syringaldehyde, vanillin, and ferulic acid. Further degradation of these aromatics sequentially produces organic acids, $CO_2$, and water. Ammonification of proteins during composting produces organic acids, aldehydes, and alcohols which also undergo subsequent oxidation to $CO_2$ and water. Fels et al. [6] described composting as a degradation reaction of carbohydrates, lipids, proteins, cellulosic, tannins, and pigments leading to the formation of smaller molecules which, later, microbial activity polymerizes and repolymerizes, resulting in the formation of humic substances. Composted material is usually used as soil fertilizer with several advantages when compared with inorganic fertilizers [7].

Through a life cycle analysis, Kim and Kim [8] showed that composting food waste has a great environmental advantage because it significantly reduces greenhouse gas emissions. According to the authors, 1 ton of food waste generates the equivalent of 1010 kg of $CO_2$ when landfilled, while composting only generates the equivalent of 123 kg of $CO_2$.

Composting has a significant effect on the thermal degradation behavior of biomass, with a marked increase in the rate of thermal degradation of the lignocellulosic components and a corresponding shift in degradation maxima to lower temperatures. The mechanisms of degradation of raw and composted biomass are analogous.

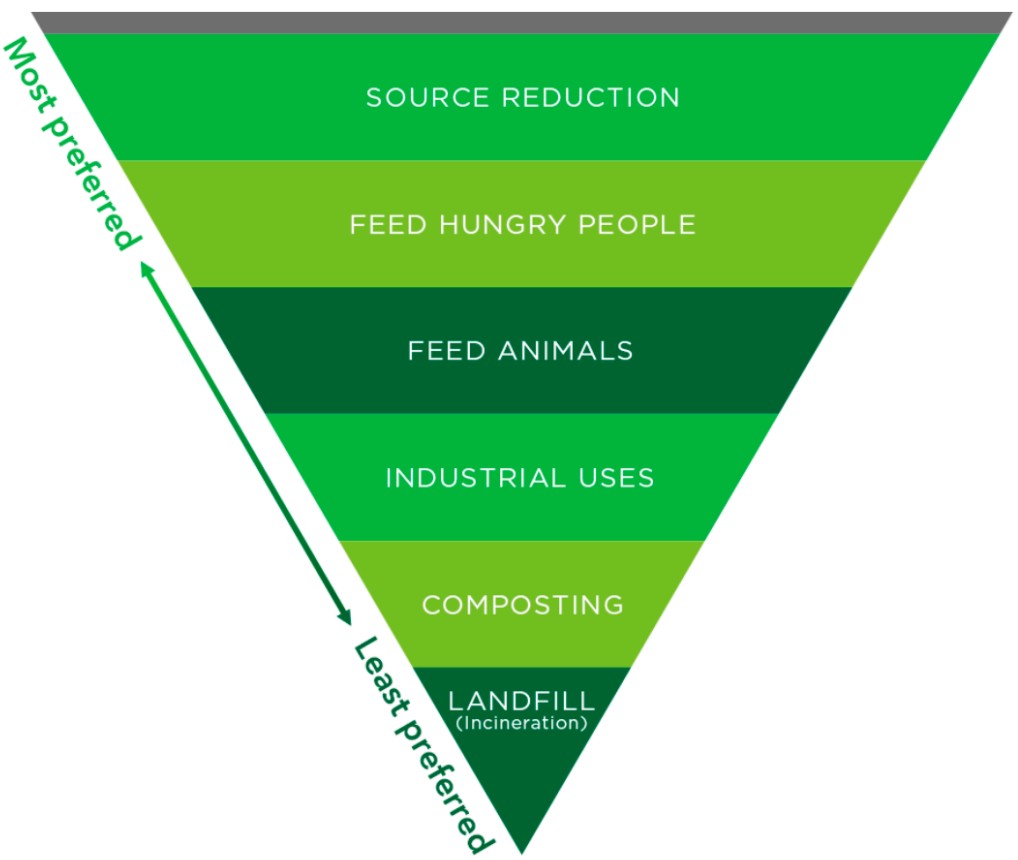

**Figure 2.** Food waste management hierarchy (adapted from [9]).

Dry, biodegraded Municipal Solid Waste (MSW), which contains a large amount of food waste [10], is a carbon-neutral source for the production of biofuels. The pyrolysis of these materials leads to the production of $H_2$-rich biogas, a bio-oil with high phenol content and mesoporous biochar [11]. The bio-oil yield is, however, lower than that obtained for non-degraded biomass, which may be due to the higher mineralization of the biodegraded material that acts as a gasification catalyst. In biomass pyrolysis, the $H_2$ is produced by volatilization of lignocellulosic components and by charring, with charring being the most effective process. For non-composted biomass, the $H_2$ from charring is about 3 times the $H_2$ from volatilization, and for composted biomass, charring is even more effective (factor greater than 3) [12]. Composted biomass produces more hydrogen due to its increased lignin content and mineralization [13]. Inorganics act as gasification catalysts [14]. Palma et al. studied the influence of composting conditions, aeration, and humidity, on the pyrolysis behavior of composted MSW. They observed maximum $H_2$ production for MSW composted in middle conditions (40% of humidity, and 50 mL/(kg.day) of aeration), whereas the biodegradability was enhanced for 55% of humidity and 175 mL/(kg.day) of aeration [15]. The authors also reported a decrease in the pyrolysis activation energy for composted MSW when compared with that of raw MSW. Juchelková et al. [16] also observed increased syngas yield for composted perennial grass. The behavior of composted grass was attributed to the biochemical components formed during composting, such as humic substances, and to improved charring reactions [17]. López-González et al. [18] attributed the improved $H_2$ formation during pyrolysis to the high mineralization of composted biomass. According to the authors, Ca is responsible for the improved $H_2$ formation, whereas K improves $CH_4$ formation.

**Figure 3.** Cellulose decomposition during composting (adapted from [17]).

Composting household garbage is a viable procedure for converting organic matter into high-value humic compounds (Figure 4), and it can be used as biomass to produce biofuel through gasification and pyrolysis, according to the published literature. To assess its potential as a third-generation biomass, the sections below offer data on the characterization of composted biomass made up of vegetable and fruit waste as well as garden waste that has been matured for 6 months.

**Figure 4.** Fluvic and humic acids molecules.

## 2. Experimental Section

### 2.1. Materials and Methods

The studies reported below were performed with biomass composted in a home compost bin (Figure 5) provided by the Department of the Environment of the local city council (Oeiras, Portugal). The composted biomass was made from kitchen leftovers that included overripe fruit, fruit peelings, vegetable scraps, and yard trash from trimming shrubs and lawns. The plants' branches had already been shredded. The materials were accumulated during a period of six months, encompassing the spring and summer seasons. Composting took place in the dark and with intermittent additions of water (open box on

rainy days). The layer of the composting box bottom that was in contact with the soil was where the biomass was collected.

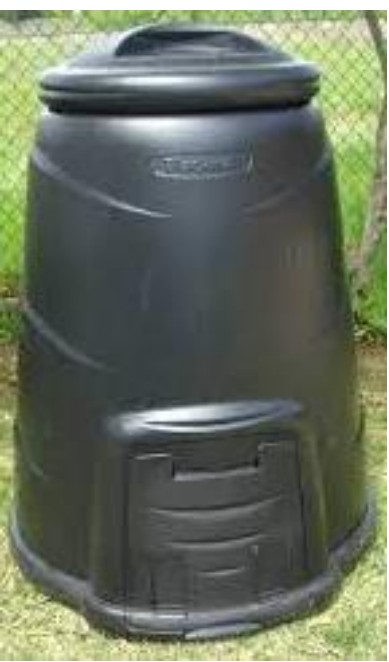

**Figure 5.** Home compost bin offered by the Oeiras City Council, Portugal (330 L).

The collected, composted biomass was dried in contact with air and by exposure to the sun for one week. Then, it was reduced to powder in a blade mill. The sample was sieved and the fraction above 350 μm was rejected. The sample was divided into three fractions and one of them was characterized without any additional treatment (fraction #1). The fraction designated as fraction #2 was treated with a 1 mol/L NaOH solution for 2 h at boiling temperature. Fraction #3 was treated with a 1 mol/L solution of $H_2SO_4$ at boiling temperature for 2 h. Both acid and basic treatment were accomplished using a solid/liquid ratio equal to 5 g/100 mL and under vigorous stirring. The solids were separated by filtration after the contact time, and then, they were left to naturally dry in contact with atmospheric air. The liquid fractions recovered by filtration resulting from the acid and basic treatments were dried in a rotavapor at 80 °C and 120 mm Hg. The brownish liquids obtained were characterized by ATR-FTIR. The spectra were acquired with a PerkinElmer Spectrum Two IR spectrometer, equipped with a Pike ATR accessory and a diamond crystal, with a resolution of 4 cm$^{-1}$, 4 scans, and a wavelength in the range of 4000–600 cm$^{-1}$. The IR spectra of solids were acquired in analogous conditions. FTIR spectra were corrected using Kubelka–Munk function to increase the signal/noise ratio [19].

The elemental composition of dried compost material was evaluated using a Micro Elemental Analyzer CHNS-O (EMA502). The total carbon, nitrogen, hydrogen, and sulfur were measured by fast combustion of a 5 mg sample at 1030 °C, in the presence of pure oxygen, and using helium as a carrier gas. The oxygen content of the sample was assessed by mass balance deducting the C and O contribution of the calcite that compose the ash. Since the ash is not calcite only, its content was considered equal to the crystalline phase fraction determined by XRD. The ash, inorganics, in the compost biomass was assessed by a classic methodology involving the dry oxidation of organic matter in a muffle at 600 °C for 30 min. The methodology was adapted from ASTM E1755-01(2020) [20].

The raw compost was characterized by microscopy. Observations were made using a stereomicroscope and a digital camera (Stereomicroscope NIKON SMZZ645 coupled with digital camera MOTICAM 10 MP). The material was also examined by SEM-EDS. Before analysis, the powdered sample was studded over a double-face carbon tape and covered by a thin film of Pd-Au. The micrographs were collected using a Hitachi S-2400 Scanning

Electron Microscope. The elemental analysis during image acquisition was performed using a Bruker light elements EDS detector at 20.0 kV.

The inorganics in composted materials were analyzed by XRD on a Burke D8 Advance X-ray diffractometer with Cu $K_a$ radiation at 40 kV and 40 mA. The diffractograms were acquired in the range of 5–70° with a step of 0.02° and 1 s by step. XRD was also used to characterize the compost ashes obtained at 600 °C and the solids obtained through crystallization of the salts leached in hot water (60 °C) soaking the compost.

The dried solid residues were characterized by thermogravimetry (TG) under $N_2$ flow to simulate the pyrolysis process. Thermogravimetry tests were performed on a Netzsch STA 490 PC thermobalance using five heating rates: 10 K/min, 20 K/min, 30 K/min, 40 K/min, and 50 K/min. About 60–100 mg of powdered sample (<750 μm) was used in each test and placed in a 100 μL alumina crucible. The nitrogen flow used during the TG tests was 18 L/h, to create an inert atmosphere in the furnace. The tests were carried out between 303 and 1373 K. The mass loss rate (DTG) was calculated using the equipment software (Proteus).

### 2.2. Estimation of the Biomass Composition and Kinetic Studies

The lignocellulosic composition of each analyzed sample was estimated from the deconvolution of the DTG curves using Gaussian curves. Details on the deconvolution procedure are provided elsewhere [21,22].

The calculated total DTG curve is the sum of the different theoretical individual curves of the different components. This deconvolution method assumes that the thermal degradation of the different components of the biomass sample occurs individually. The fit of the experimental DTG, using symmetric Gaussian curves, is performed by the optimization of three parameters: $a_i$, amplitude; $b_i$, position; $c_i$, width at half the maximum height, T indicates the temperature values (in K), and i refers to the individual component, according to Equation (1) [21,22].

$$DTG_{theo} = \sum_i a_i \exp\left[ -\left( \frac{T - b_i}{c_i} \right)^2 \right] \tag{1}$$

To minimize the difference between the total calculated curve (with the different individual curves of the Gaussian type) and the experimental curve of the DTG, the parameters $a_i$, $b_i$, and $c_i$ of Equation (1) were fitted using the least-squares method.

$$Opt. = \sum_T \left( \frac{DTG - DTG_{theo}}{DTG} \right)^2 \tag{2}$$

This objective function (O. F.), Equation (2), was performed using Microsoft Excel Solver. This data treatment is described in more detail in previous works [21,22].

The kinetic parameters, in particular the activation energy and the pre-exponential factor, were computed using the Kissinger, model-free, method from thermogravimetric experimental data obtained under nitrogen flow at different heating rates according to Equation (3) [21,22].

$$\ln\left( \frac{\beta}{T_m^2} \right) = \ln\left( \frac{AR}{E_a} \right) - \frac{E_a}{RT_m} \tag{3}$$

where $T_m$ is the temperature corresponding to the maximum rate of thermal degradation, $\beta$ is the heating rate (K/min), $E_a$ is the activation energy, expressed in kJ/mol, R is the ideal gas constant, and A is the pre-exponential factor ($s^{-1}$).

## 3. Results and Discussion

### 3.1. Biomass Characterization

After drying, the compost material had a heterogeneous look (Figure 6) with small fragments of hollowed-out shrub trunks still visible (a), but with a significant degree of degradation. Following grinding, a less dense, more uniform, and lighter, brown-colored

powder than the raw material was obtained (b). The homogeneity is only macroscopic because the visualization of the compost under a stereomicroscope (c) underlines its heterogeneity, with visible fibrous materials (resilient non-degraded lignocellulosic components) and rounded shiny and translucent materials that correspond to the inorganic components.

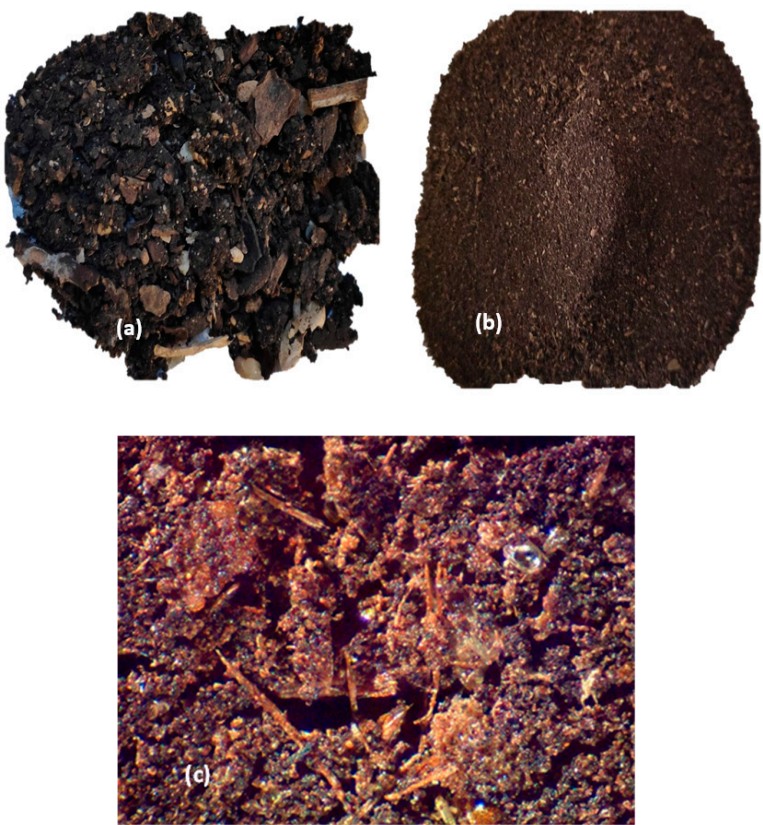

**Figure 6.** Raw (**a**) and ground (**b**) composted biomass (previously dried) and ground material observed by stereomicroscope (image with 2.5 mm) (**c**).

Overall, the composted biomass sample exhibits high carbon and oxygen content, but a low percentage of hydrogen, nitrogen, and sulfur as shown in Table 1. The ash content is high, about 37.6%. Barneto et al. reported that composted biomass has a higher ash content compared to the original raw material [13]. In particular, Barneto et al. found that the new components of the stable organic matter (compost) have different thermal behavior than the original biomass [13]. Thus, a likely reason for this high ash content is due to the high thermal stability of the new composts causing thermal decomposition at high temperatures and the presence of inorganic materials. The H/C ratio is within the range for biomass samples, but the O/C ratio is lower than expected [23]. The decrease in the O/C ratio is to be expected since, during the composting process, bacteria, fungi, and other microorganisms decompose the raw material by consuming oxygen and releasing heat, water, and carbon dioxide [13].

**Table 1.** Carbon, hydrogen, nitrogen, sulfur, oxygen, ash content (%), H/C, and O/C ratios of the composted biomass.

| | C | H | N | S | Ash | O ★ | Atomic Ratio | |
| --- | --- | --- | --- | --- | --- | --- | --- | --- |
| | | | | | | | H/C | O/C |
| | | | | (Wt. %) | | | | |
| Composted Biomass | 35.4 ± 0.7 | 4.7 ± 0.2 | 2.7 ± 0.3 | 0.3 ± 0.4 | 37.6 | 19.3 | 1.6 | 0.4 |

★ assessed by difference.

*3.2. Infrared Spectroscopy*

3.2.1. Liquid Fraction from Treatments

To understand the changes in the structural properties of the raw compost before and after the acid and alkaline pretreatments, FTIR analyses were performed (Figure 7). The liquids obtained from the acid and alkaline treatments of the samples exhibited different spectra between 1800 and 700 cm$^{-1}$ and very similar bands in the absorption region at 3700–2900 cm$^{-1}$, which is related to the single bond -OH stretching of phenolic hydroxyl and aliphatic hydroxyl groups. In the region of 3000–2800 cm$^{-1}$, the bands were mainly due to C-H stretching of methyl and methylene groups. Bands in the range of 1200–1000 cm$^{-1}$ describe the existence of C-O bond stretching. This indicates that the acid pretreatment broke the bonds of complex lignin structures, as reported by [24]. The band between 3700 and 2900 cm$^{-1}$ for acid-treated samples weakened, indicating that the amount of -OH groups decreased in comparison to the alkaline-treated samples. These results showed that the acid pretreatment conditions exert more influence on broken C-O bonds, while -OH groups are more easily broken during alkaline pretreatment.

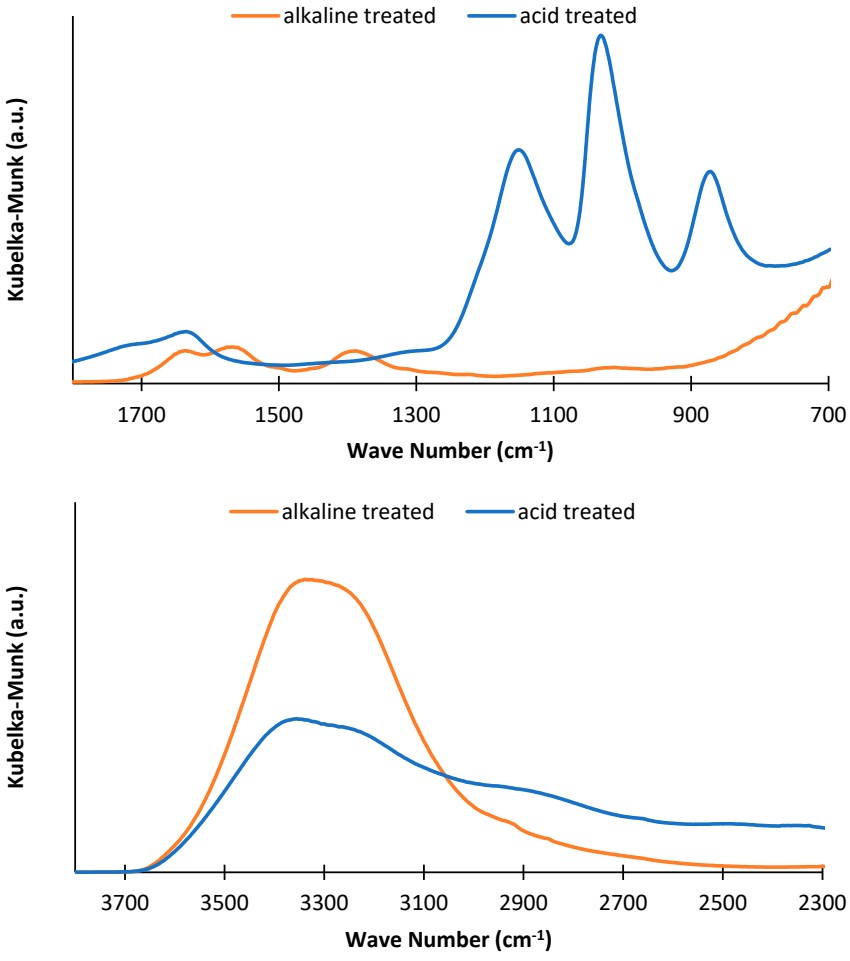

**Figure 7.** FTIR spectra of the liquid fraction of treatments.

3.2.2. Solid Fractions after Treatments

The FTIR bands of dried solids, raw compost, and leftovers from H$_2$SO$_4$ and NaOH treatments (Figure 8) were analyzed using data from the literature summarized in Tables S1 and S2 (supplementary data section) for lignocellulosic and humic and fulvic materials. The raw compost FTIR spectrum clearly shows two bands of humic and fulvic compounds in the 3700–3000 cm$^{-1}$ region. The band in the 3700–3620 cm$^{-1}$ range belongs to free OH, while the band in the 3400–3200 cm$^{-1}$ range belongs to OH, NH, and phenol OH vibration

modes, according to the data in Table S2. A low intensity and broad feature in the range of 1700–1500 cm$^{-1}$ belongs to amide, quinone, and ketone, as well as to aromatic C=C of humic and fulvic substances [25]. Lignin also presents C=O vibration mode in the 1700–1500 cm$^{-1}$ range (Table S1). The C=O band around 1737 cm$^{-1}$ is absent, which is compatible with the compost maturity (3 months). FTIR bands of humic and fulvic substances overlap the bands of lignocellulosic components since they are the major components. The spectra of the leftovers (biomass after acid and basic treatments) show a decrease in band intensity in the 3700–3000 cm$^{-1}$ region, which can be attributed to the leaching of the humic and fulvic substances. For the alkali leftovers, the band is centered around 3300 cm$^{-1}$ which is attributable to OH groups of lignin [26]. The spectrum of the acid leftovers shows an intense band centered around 1400 cm$^{-1}$ which can be attributed to lignin [27] overlapped with Si-O vibration [28] mode from quartz. Calcite also presents a vibration band that can overlap in the 1500–1330 cm$^{-1}$ range [29]. The band at 871 cm$^{-1}$ also belongs to calcite [29]. The spectrum of the basic solid leftovers mainly presents in the 1700–700 cm$^{-1}$ range, with a pair of bands at 1099 cm$^{-1}$ and 1041 cm$^{-1}$ attributable to cellulose since the NaOH treatment of woody materials led to delignification [30]. Comparing the spectra of basic and acid solid leftovers, it can be concluded that basic treatment also promotes the leaching of inorganics such as Si and Ca compounds.

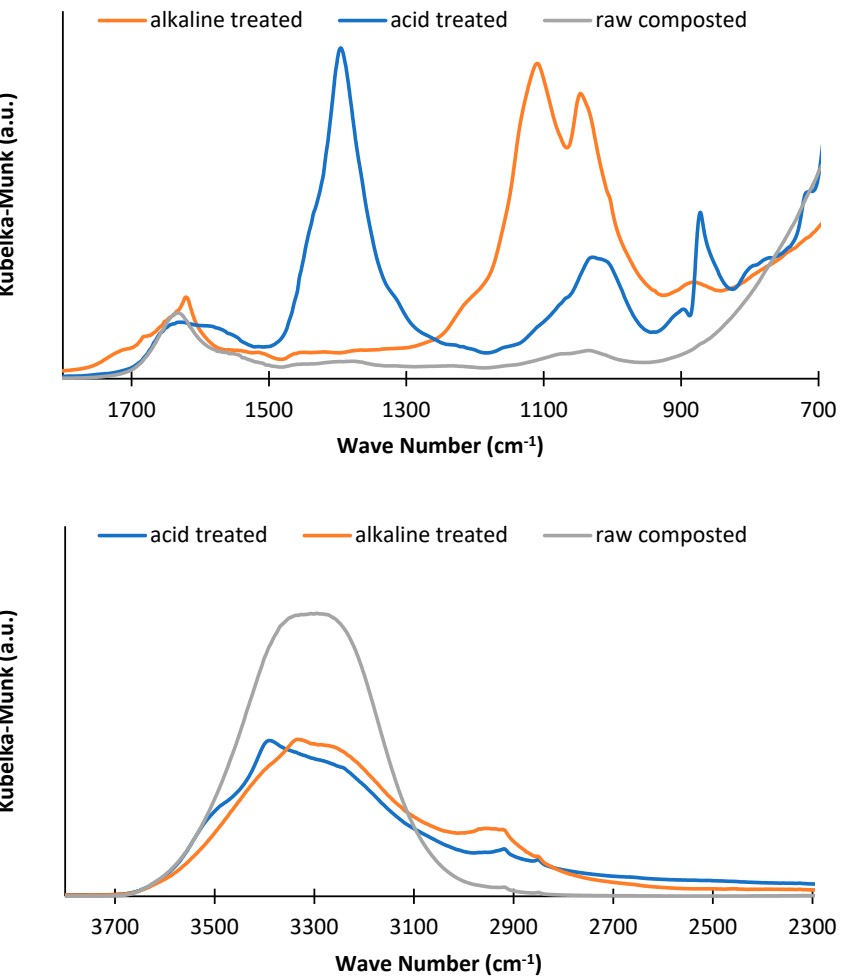

**Figure 8.** FTIR spectra of solid fractions, raw compost, and leftovers from H$_2$SO$_4$ and NaOH treatments.

### 3.3. SEM—EDS

Analysis of the morphological surfaces of the compost using SEM-EDS showed that there is heterogeneity in the size and shape of the sample particles, as presented in Figure 9. The compost, which is a mixture of various kitchen leftovers, shows a morphologically

irregular shape. The micrograph (c) in Figure 9 shows a morphology similar to nano cellulose [31]. Nanocellulose is obtained from the derivatization of cellulose by delignification and alkaline and acid hydrolysis [30]. Additionally, the specific morphology in the micrograph c can also be defined as the characteristic soil flora, as reported by [32]. Slow decomposition processes under bacterial action may have favored the appearance of soil flora. According to Czarnecka-Komorowska et al. [32], the typical soil flora was found in the microbial tests of biocomposites carried out after composting in a pile.

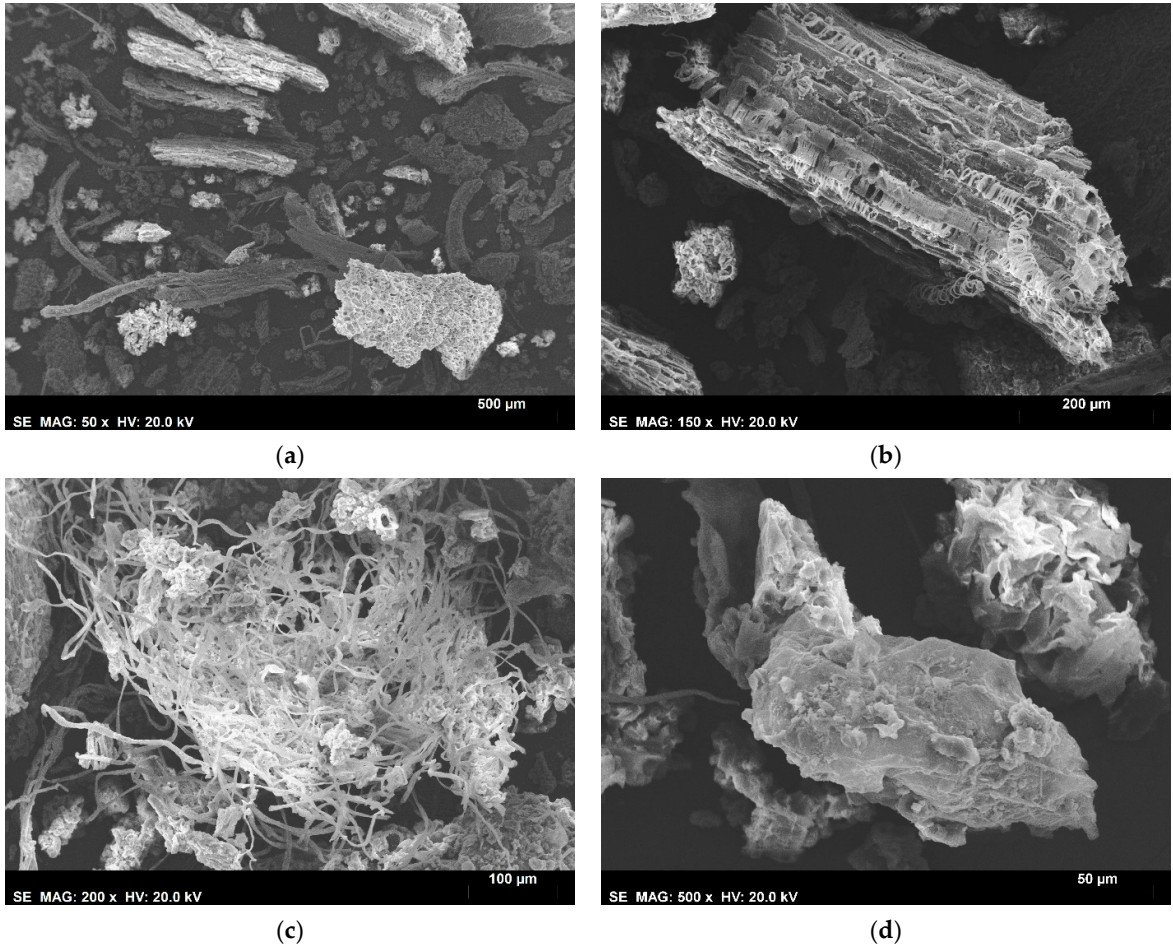

**Figure 9.** SEM micrographs of composted biomass.

Figure 10 shows that the main components present in the composted material were C and O. This is to be expected since the residues, being mainly composed of sugars, have functional groups which contain C-O and O-H bonds.

X-ray diffraction was used to examine the compound's inorganic crystalline phases. The diffractograms of the soluble salts, the composite following salt solubilization, and the ash produced at 600 °C are shown in Figure 11. The crystalline phases are identified in Figure S1 of the supplementary data section. The composted biomass contains water-soluble potassium salts, which crystallize as sylvite (KCl) after evaporation. EDS analysis performed during the capture of SEM micrographs (Figure 10) corroborated the existence of K. After leaching with water, the composite material shows diffraction stripes from quartz ($SiO_2$), calcite ($CaCO_3$), and, to a lesser extent, whewellite ($CaC_2O_4 \cdot H_2O$). The quartz diffraction pattern dominates the diffractogram for the ashes collected at 600 °C, with low-intensity diffraction patterns of calcite ($CaCO_3$), aragonite ($CaCO_3$), halite (NaCl), illite, and albite also present. The latter two crystalline phases contain aluminum, which was found in the SEM-EDS investigations (Figure 10). The composition of the ash obtained at 600 °C is comparable to catalysts reported in the literature for $H_2$ production by biomass

gasification [33], signaling a potential application for the ash if the compost is employed for energy production by combustion. The other inorganics detected by EDS (Figure 10) or suggested by stereomicroscopy observations in the compost sample were below the XRD detection limit or well dispersed in the biomass, in amorphous or nanosized crystals, thus being undetectable. Data for the characterization of inorganics in composted biomass are essentially non-existent in the literature, and the availability of inorganics in compost will depend on the biomass mixture employed, making the comparison with published data even more difficult.

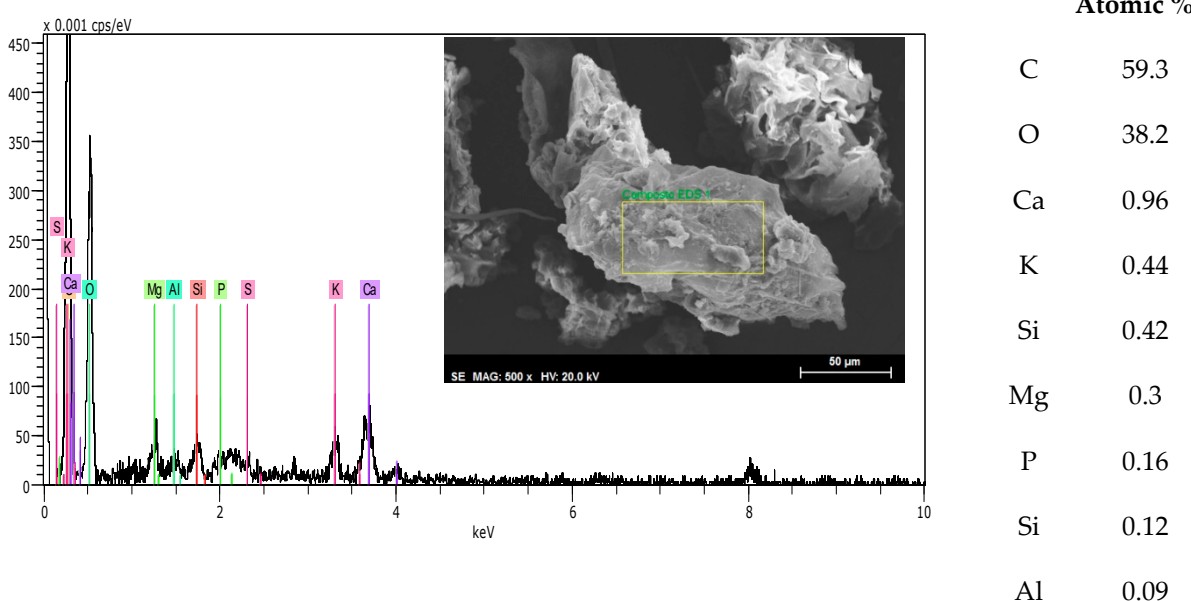

| | Atomic % |
|----|----|
| C | 59.3 |
| O | 38.2 |
| Ca | 0.96 |
| K | 0.44 |
| Si | 0.42 |
| Mg | 0.3 |
| P | 0.16 |
| Si | 0.12 |
| Al | 0.09 |

**Figure 10.** EDS analysis of composted material (overlapped micrograph shows the analyzed surface).

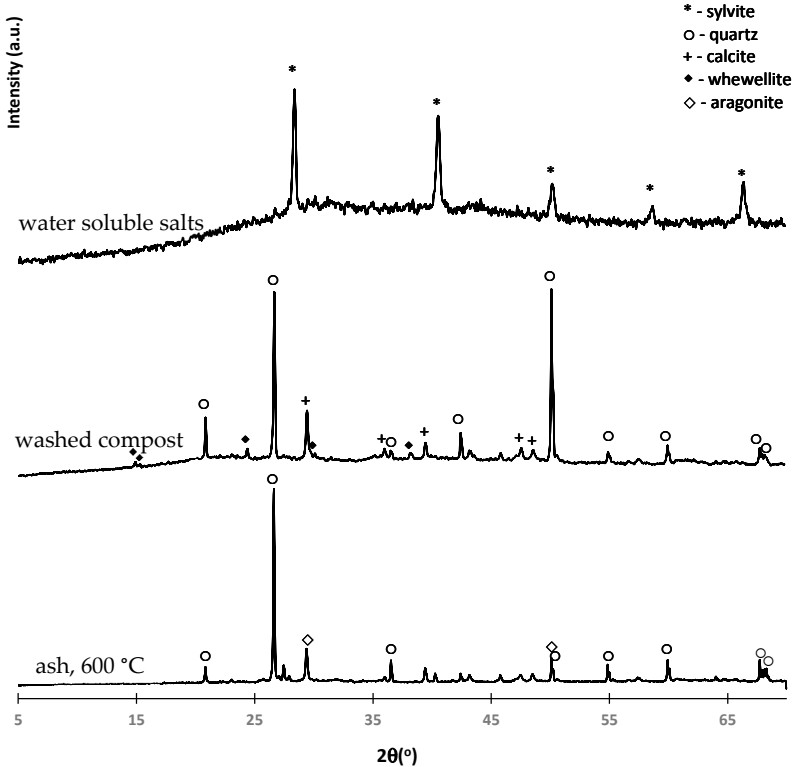

**Figure 11.** XRD of ashes obtained at 600 °C, soluble salts, and washed compost. Phase attribution in Figure S1 in the supplementary data section.

### 3.4. Thermogravimetry Tests

To study the impact of the acid and alkaline treatment on the structure of the composted biomass sample, thermogravimetry tests were carried out. Figure 12 shows the impact of each pretreatment on each component such as moisture, volatile extractive, hemicellulose, cellulose, and lignin.

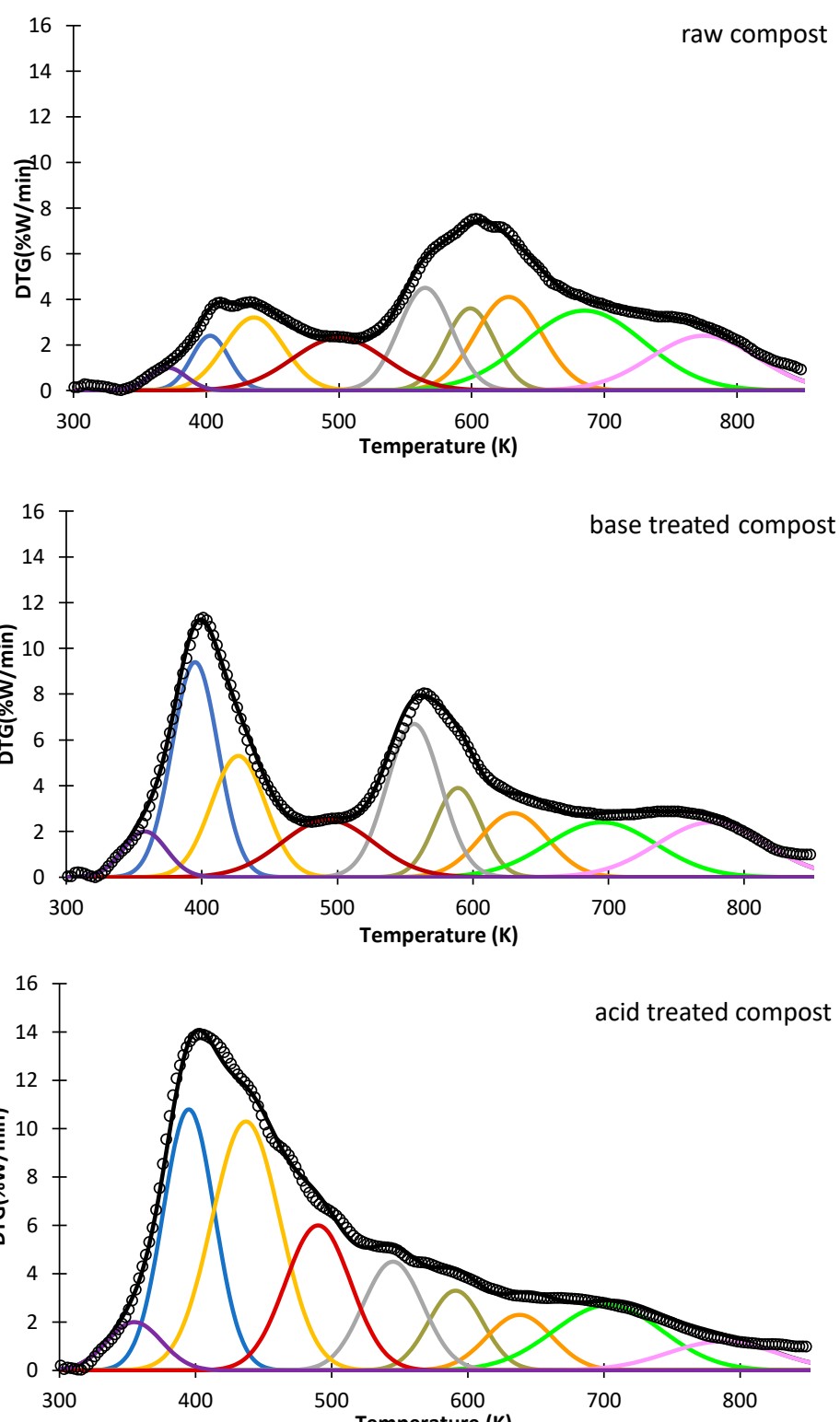

**Figure 12.** DTG experimental (○) and computed (-) by Gauss curve deconvolution of raw and acid- and base-treated biomasses (40 °C/min; N$_2$ flow).

Observing the DTG profiles of the samples treated with acidic and base solutions, it is verified that there is a greater thermal degradation of these samples at temperatures between 300 and 500 K compared to the raw compost sample. At this temperature range, it is considered that there is volatilization of light compounds, extractives, and water evaporation. This shows that both treatments chemically changed the structures of the samples to make them easier to thermally degrade. From Table 2, it is possible to observe that, in the treated samples, the amount of all components, except moisture and volatile extractives, decreased concerning the raw compost. The presence of high amounts of water in the treated samples, around values between 17 and 20%, is expected since the samples were dried in atmospheric air.

**Table 2.** Weight composition of raw and treated compost.

| Compost | Weight Composition (%) | | | | | | |
|---|---|---|---|---|---|---|---|
| | Moisture | Extractive Volatile | Component 1 | Component 2 | Component 3 | Lignin 1 | Lignin 2 |
| Raw | 4.7 | 23.4 | 12.6 | 9.4 | 14.4 | 22.1 | 13.5 |
| Base-treated | 17.5 | 28.4 | 15.6 | 6.3 | 8.8 | 11.6 | 11.8 |
| Acid-treated | 20.0 | 43.2 | 9.7 | 6.5 | 5.4 | 10.6 | 4.7 |

Carvalho et al. reported that alkaline pretreatments using eucalyptus biomass generated the production of fragments of lignin degradation products and polysaccharide degradation products that led to the formation of pseudo-extractives [34]. This observation agrees with the results obtained.

Komilis et al. [35] observed that hemicellulose was practically entirely degraded in all substrates studied while cellulose degradation ranged from 53.9to 91.1% for all substrates. Lignin was the least degraded during the tests. The maximum duration of experimental runs was close to six months. Based on this study, it is possible to approximate Komilis' results with those obtained in this paper considering that hemicellulose is already highly degraded, and cellulose and lignin are partially degraded. Although the raw compost sample has already undergone biological degradation, it can be assumed that component 1 could refer to the degraded structure of hemicellulose while components 2 and 3 to cellulose. The values obtained for the weight composition of the composted biomass are within the range of values reported by Komilis et al. [35].

Subhedar et al. reported that delignification, in an alkaline medium, increases with increasing temperature. The increase in temperature from 313 to 373 K increased the degree of delignification from 16% to 44% [36]. From the data in Table 2, considering the ratio of the percent of total lignin of the alkaline-treated sample with the percent of the composted biomass, it can be observed that about 44% suffered delignification.

These results follow what has been described in the literature since the acid pretreatment of lignocellulosic biomass is based on breaking the glycosidic bonds of hemicellulose and cellulose into sugar monomers, while alkali pretreatment results in the solubilization of hemicellulose and lignin fragments in the alkali solution [37]. However, from Table 2, it appears that both acidic and alkaline treatments produced lignin solubilizing effects in the respective solutions.

In general, acid treatment causes greater degradation/solubilization of all components than alkaline treatment. It has been reported that diluted acid pretreatment was a more feasible method for wild rice grass compared to alkaline pretreatment [38,39].

Regarding the kinetic results present in Table 3, it can be seen that the energy activation of component 1 and component 2 using acid treatment shows the smallest Ea compared to the raw material compost or alkaline-treated compost. This can be beneficial for energy production because the smaller the activation energy is, the smaller the amount of energy needed for the conversion process of the component.

**Table 3.** Kinetic parameters obtained by the Kissinger method.

| | Raw Compost | | | Base-Treated Compost | | | Acid-Treated Compost | | |
|---|---|---|---|---|---|---|---|---|---|
| | Tm (K) | Ea (kJ mol) | k (s$^{-1}$) | Tm (K) | Ea (kJ mol) | k (s$^{-1}$) | Tm (K) | Ea (kJ mol) | k (s$^{-1}$) |
| Component 1 | 559 | 201 | $1 \times 10^{33}$ | 551 | 132 | $8 \times 10^{36}$ | 542 | 187 | $3 \times 10^{45}$ |
| Component 2 | 595 | 288 | $1 \times 10^{31}$ | 581 | 167 | $5 \times 10^{49}$ | 586 | 212 | $1 \times 10^{57}$ |
| Component 3 | 623 | 307 | $3 \times 10^{27}$ | 626 | 345 | $5 \times 10^{32}$ | 634 | 300 | $3 \times 10^{51}$ |
| Lignin 1 | 679 | 250 | $1 \times 10^{51}$ | 692 | 548 | $5 \times 10^{98}$ | 700 | 484 | $4 \times 10^{48}$ |
| Lignin 2 | 767 | 198 | $2 \times 10^{32}$ | 772 | 471 | $9 \times 10^{45}$ | 783 | 457 | $7 \times 10^{30}$ |

Carvalho et al. observed that lignin and xylan fragments fractionated during pretreatments acted as inhibitors during enzymatic hydrolysis [34]. This may show that solids remaining after pretreatments can be difficult to degrade. This corroborates with the results in which high activation energy values were obtained for compounds that were part of the lignin composition. The more difficult the degradation, the higher the observed activation energy.

**4. Conclusions**

The management of domestic organic waste, both food and garden waste, is a current issue with substantial economic expenses connected with minimizing the environmental problems associated with its mishandling. The vermicomposting of these leftovers is suggested as a potential methodology for organic waste treatment and management because it does not require complicated equipment and is not energy intensive. The compost produced can be used as fertilizer, but the energy requirements of modern society, along with the properties of the stabilized organic matter, prompt research on the possibility of using it as biomass for the generation of biofuels. The kinetic data acquired by thermogravimetry reveal that the composted biomass comprises distinct fractions with pyrolysis activation energy in the range of 200–300 kJ/mol, boosting $H_2$ production during pyrolysis, making the process more energy efficient according to the literature. The acid and basic treatments raise the pyrolysis activation energy of the resulting solids, making them ideal for the synthesis of carbonaceous materials. Because of the compost's high inorganic content and the fact that it remains in the biochar created, the biochar obtained will be suitable for noble applications such as electrochemistry. Future research on the use of composted biomass for the gasification process that produces hydrogen is pertinent given the significance that hydrogen has taken on in modern society as a clean fuel. The inorganic components of the biomass, ashes, will serve as catalysts for this thermochemical process.

**Supplementary Materials:** The following supporting information can be downloaded at https://www.mdpi.com/article/10.3390/reactions4020018/s1, Figure S1: X-ray diffractograms with crystalline phases identification; Table S1: FTIR bands of lignocellulosic (adapted from [40]); Table S2: FTIR bands attribution for humic and fulvic substances in soils (adapted from [25]). References [25,40] are cited in the Supplementary Materials.

**Author Contributions:** A.P.S.D.—conceptualization, supervision, writing—review and editing; B.R.—supervision, formal analysis, writing—review and editing; N.D.S.—investigation; R.G.d.S.—resources, investigation; M.F.C.P.—resources, investigation; O.F.—investigation; R.Z.—formal analysis, review; O.D.P.—formal analysis, review. All authors have read and agreed to the published version of the manuscript.

**Funding:** This research was supported by the Fundação para a Ciência e a Tecnologia through CERENA, under project UID/ECI/04028/2019.

**Informed Consent Statement:** Not applicable.

**Data Availability Statement:** Data is contained within the article and Supplementary Material.

**Acknowledgments:** This publication is based upon work from COST Action Waste biorefinery technologies for accelerating sustainable energy processes (WIRE), CA20127, supported by COST (European Cooperation in Science and Technology); www.cost.eu.

**Conflicts of Interest:** The authors declare that they have no known competing financial interest or personal relationship that could have appeared to influence the work reported in this paper.

## Abbreviations

| | |
|---|---|
| A | Pre-exponential factor of Arrhenius equation $(s^{-1})$ |
| ATR | Attenuated total reflectance |
| DTG | Rate of thermal degradation (%wt./min) |
| Ea | Activation energy (kJ/mol) |
| EDS | Energy dispersive X-ray spectroscopy |
| FTIR | Fourier transform infrared spectroscopy |
| GDP | Gross domestic product |
| MSW | Municipal solid waste |
| R | Ideal gas constant (J/(mol.K) |
| SEM | Scanning electron microscopy |
| TG | Thermogravimetry |
| Tm | Temperature corresponding to the maximum rate of thermal degradation (K) |
| XRD | X-ray diffraction |
| β | Heating rate (K/min) |

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
