# Peer review of "Biofuels from Pyrolysis of Third-Generation Biomass from Household and Garden Waste Composting Bin: Kinetics Analysis"

_reactions, doi:10.3390/reactions4020018_

Round 1

Reviewer 1 Report

A brief English editing should be carried out in the whole text.

Reviewer 2 Report

The author may revise the title with informative keywords with catchy.

The author may justify the reason behind this vermicompost of household study

Correct it "H2SO4 and H2"

Rephrase this sentence "A 6-month (spring and summer seasons) matured compost containing vegetable and fruit scraps from household litter and grass and shrub trimmings from yard waste was characterized to evaluate its viability as a raw material for pyrolysis processes"

The aim, highlight, and conclusion part of the abstract need to be improved

Change the keywords

Rephrase the sentence "Today’s society generates massive amounts of food waste, which must be handled 26 and managed to reduce environmental impacts."

mention the examples of greenhouse gases

Provide appropriate reference for this sentence "Moreover, the world population will reach 10 bil-33 lion by 2050, and under current conditions, it will be impossible to produce food to feed 34 the entire population"

change the figure 1 pattern for ease and clear information of  the figure.

Change the color of words mentioned in the figure 2 

include the country name "Department of Environment of the local city council"

incorporate the model and manufacturer of scientific equipment "e.g. ATR-FTIR." used.

The author should mention the FTIR peaks in all FTIR figure

The author should provide a full form for acronyms at the time of the first entry

The manuscript format needs to be checked and changed as per the journal format

The author should check the entire manuscript to correct the typographical grammaticall errors.

Future perspectives need to be added in the conclusion section.

Check the uniformity in font size and style in the entire manuscript

Improve the conclusion section and suggests the possible future perspectives

The author should correct the table format and increase the figure resolution during the revision

Correct the reference list format as per the journal format

The manuscript need language correction

Round 2

Reviewer 1 Report

The revised manuscript can be accepted for publication

Author Response

The authors thank the reviewer for the comment.

Reviewer 2 Report

The author may justify the reason behind this vermicompost of household study

The aim, highlight, and conclusion part of the abstract need to be improved 

Improve the language of the manuscript

The authors failed to address some comments, which were given already
